# CROC: PRETRAINING LARGE MULTIMODAL MODELS WITH CROSS-MODAL COMPREHENSION

## ABSTRACT

Recent advances in Large Language Models (LLMs) have catalyzed the development of Large Multimodal Models (LMMs). However, existing research primarily focuses on tuning language and image instructions, ignoring the critical pretraining phase where models learn to process textual and visual modalities jointly. In this paper, we propose a new pretraining paradigm for LMMs to enhance the visual comprehension capabilities of LLMs by introducing a novel cross-modal comprehension stage. Specifically, we design a dynamically learnable prompt token pool and employ the Hungarian algorithm to replace part of the original visual tokens with the most relevant prompt tokens. Then, we conceptualize visual tokens as analogous to a "foreign language" for the LLMs and propose a mixed attention mechanism with bidirectional visual attention and unidirectional textual attention to comprehensively enhance the understanding of visual tokens. Meanwhile, we integrate a detailed caption generation task, leveraging rich descriptions to further facilitate LLMs in understanding visual semantic information. After pretraining on 1.5 million publicly accessible data, we present a new foundation model called **Croc**. Experimental results demonstrate that Croc achieves new state-of-the-art performance on massive vision-language benchmarks. To support reproducibility and facilitate further research, we will release the training code and pre-trained model weights.

## 1 INTRODUCTION

The rapid expansion of mobile networks has accelerated the generation of vast data volumes, presenting unprecedented opportunities for the development and application of Large Language Models (LLMs) (Zhao et al., 2023; Touvron et al., 2023; Bai et al., 2023). Despite their effectiveness, LLMs are primarily confined to processing textual inputs. To expand their multimodal perceptual capabilities, there is an increasing research focus on Large Multimodal Models (LMMs) (Yin et al., 2023; Jin et al., 2024; Yang et al., 2023b), which are designed to process and integrate inputs across multiple modalities.

As a milestone in LMM research, LLaVA (Liu et al., 2024b) leverages the language-only capabilities of GPT-4 (Achiam et al., 2023) to generate multimodal language-image instruction-following datasets, demonstrating impressive multimodal conversational capabilities. Building on this groundwork, LLaVA-1.5 (Liu et al., 2024a) enhances performance through simple modifications to the original LLaVA framework and incorporates academically oriented Visual Question Answering (VQA) datasets with structured response formatting prompts. In parallel, BLIP-2 (Li et al., 2023a) and MiniGPT-4 (Zhu et al., 2023) connect a frozen pre-trained vision encoder and a language model through a trainable Q-Former or a linear layer, effectively mapping image features into the input embedding space of the language model. Nonetheless, these methods achieve only a superficial integration of image features within the language model's embedding space. In contrast, CogVLM (Wang et al., 2023) introduces a trainable visual expert module into the attention and Feed-Forward Network (FFN) layers of the language model. Despite this innovation, the freezing of the LLM limits its capability to attain an in-depth understanding of visual features directly.

Recent research highlights the crucial role of the pretraining process in LMMs. Flamingo (Alayrac et al., 2022) synergistically integrates pre-trained vision and language models through pretraining on comprehensive multimodal web corpora that combine text and image, enabling the performance

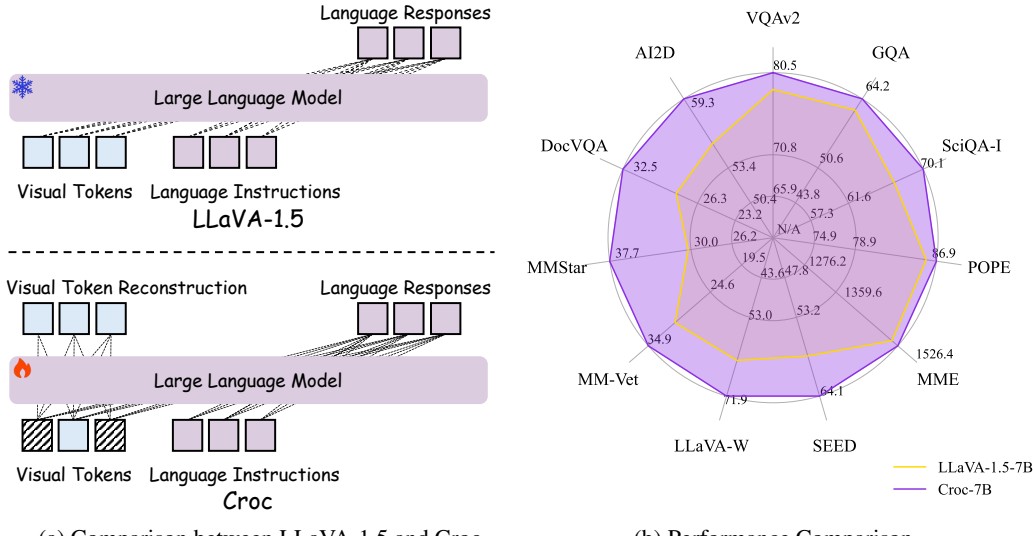

(a) Comparison between LLaVA-1.5 and Croc

(b) Performance Comparison

Figure 1: (a) Our Croc model introduces an additional pre-training stage with visual token reconstruction. (b) Our Croc model outperforms LLaVA-1.5 on a wide range of multimodal tasks.

of diverse multimodal tasks such as captioning, visual dialogue, and visual question-answering. However, the usage of large-scale pretraining data results in substantial resource consumption. VILA (Lin et al., 2024) reveals that the pretraining process substantively augments various model capabilities, including multi-image reasoning, improved in-context learning, and enriched world knowledge. Nevertheless, VILA uses 50M of the interleaved pre-training corpus to improve data diversity, which is more computationally expensive than LLaVA-1.5. LaVIT (Jin et al., 2023) introduces a meticulously designed visual tokenizer to transform non-linguistic images into a sequence of discrete tokens, thereby rendering them analogous to a foreign language that is interpretable by LLMs. However, this direct input of visual tokens into LLMs and fostering visual understanding through next-token prediction encounters significant hurdles. This limitation primarily originates from the inherent discrepancies between visual and textual tokens, particularly the absence of a robust causal linkage between sequential visual tokens. Furthermore, the application of unidirectional attention mechanisms in this context further restricts the LLM's capacity to effectively comprehend discrete visual tokens.

In this paper, we introduce a novel pretraining paradigm for LMMs designed to significantly enhance the visual comprehension capabilities of LLMs by incorporating a pioneering cross-modal comprehension stage. Specifically, we design a dynamically learnable prompt token pool and apply the Hungarian algorithm to selectively replace part of the original visual tokens with the most relevant prompt tokens. Then we conceptualize visual tokens as analogous to a "foreign language" for the LLMs and propose a mixed attention mechanism with bidirectional visual attention and unidirectional textual attention to improve the understanding of visual tokens. Meanwhile, we integrate a detailed caption generation task, leveraging rich image descriptions to further facilitate LLMs in understanding visual semantic information. Experiment results demonstrate that our proposed Croc model achieves new state-of-the-art performance across multiple benchmarks. The main contributions of this paper are summarized as follows:

- We introduce a new pretraining paradigm to enhance the visual comprehension capabilities of LLMs by introducing a novel cross-modal comprehension stage. This stage integrates visual token reconstruction and targets detailed caption generation.

- For visual token reconstruction, we design a dynamically learnable prompt token pool and employ the Hungarian algorithm to replace part of the original image tokens with the most relevant prompt tokens. In addition, we propose a mixed attention mechanism with bidirectional visual attention and unidirectional textual attention for more comprehensive visual token understanding.

- Experimental results demonstrate that our proposed Croc model achieves new state-of-the-art across various benchmarks and exhibits robust visual understanding and reasoning capabilities. We will release the training code and models to facilitate future research.

## 2 RELATED WORK

**Large Multimodal Model Pre-training.** As a significant advancement, LLaVA (Liu et al., 2024b) meticulously filters the CC3M (Changpinyo et al., 2021) dataset down to 595K and maintains the frozen state of both the visual encoder and the LLM weights, exclusively training the projection layer to align features from the visual encoder and the LLM. However, this strategy predominantly results in limited deep feature integration between the visual encoder and the LLM, primarily due to the constraints imposed by the projection layer. To address this limitation, CogVLM (Wang et al., 2023) introduces a trainable visual expert module into the attention and feed-forward network layers of the language model. Despite this innovation, constrained by the frozen state of the LLM, it continues to face challenges in comprehending the "foreign language" of visual tokens. Recently, LaVIT (Jin et al., 2023) introduces a well-designed visual tokenizer to convert non-linguistic images into a sequence of discrete tokens. However, directly inputting visual tokens into LLM to enhance visual understanding of LLM through next-token prediction presents significant limitations. VILA (Lin et al., 2024) proposes an interleaved pertraining stage to augment the LLM to support visual input, but it relies on a 50M pertraining dataset, necessitating considerable computational resources.

**Cross-Model Comprehension.** To improve the performance of Masked Image Modeling (MIM) based pre-training methods, MVP (Wei et al., 2022) initially explores the integration of multimodal pre-training within the MIM framework. Subsequently, diverging from conventional methods that predominantly predict raw pixels or low-level features, MILAN (Hou et al., 2022) adopts an innovative approach by reconstructing image features infused with substantial semantic content derived from caption supervision. UnMasked Teacher (Li et al., 2023b) selectively masks video tokens exhibiting low semantic content and aligns the remaining unmasked tokens through a linear projection to their counterparts from the teacher model. Experimental results confirm that this approach achieves state-of-the-art performance across various video-related tasks. In a recent study, RILS (Yang et al., 2023a) introduces a novel pre-training framework that employs masked visual reconstruction within a language semantic space. This framework facilitates the extraction of structured information by vision models through the accurate semantic prediction of masked tokens. Meanwhile, EVA (Fang et al., 2023) demonstrates that recovering the masked-out tokenized semantic vision features is an efficient strategy for vision-centric representation learning, obviating the need for semantic feature quantization or further tokenization. Inspired by the above works, we propose a visual token reconstruction task to improve the visual comprehension capability of LLMs.

## 3 METHODOLOGY

### 3.1 PRELIMINARIES OF LLAVA AND LLAVA-1.5

As the seminal work of visual instruction tuning, LLaVA (Liu et al., 2024b) presents the first attempt to use language-only GPT-4 (Achiam et al., 2023) to generate multimodal language-image instruction-following data. The framework of LLaVA comprises three essential components: a Visual Encoder for transforming input images into distinct visual embeddings, a Projector for mapping visual embeddings into the textual embedding space, and a Large Language Model for processing both visual and textual tokens and generating corresponding responses. LLaVA utilizes a two-stage instruction-tuning process. In the first stage, image-text pairs are converted to the single-turn conversation which requests the assistant to describe the image. Given the input visual tokens $T_v$ and textual tokens $T_t$, both $T_v$ and $T_t$ are fed into LLM to produce a coherent response. The ground-truth prediction answer is represented by the original caption $T_c$. For a sequence of length $L$, the probability of generating contextually original caption $T_c = \{c_i\}_{i=1}^{L}$ is calculated as follows:

$$p(T_c|T_v, T_t) = \prod_{i=1}^{L} p(c_i|T_v, T_{t,<i}, T_{c,<i}). \tag{1}$$

In the first stage, both the visual encoder and LLM weights are frozen and only the projection layer is updated. In the second stage, LLaVA keeps the visual encoder weights frozen and updates both the pre-trained weights of the projection layer and the LLM. With simple modifications to LLaVA, LLaVA-1.5 (Liu et al., 2024a) integrates CLIP ViT-L/14@336px (Radford et al., 2021) with an MLP projection and incorporates academic task-oriented VQA data with response formatting prompts, resulting in better multimodal comprehension capability.

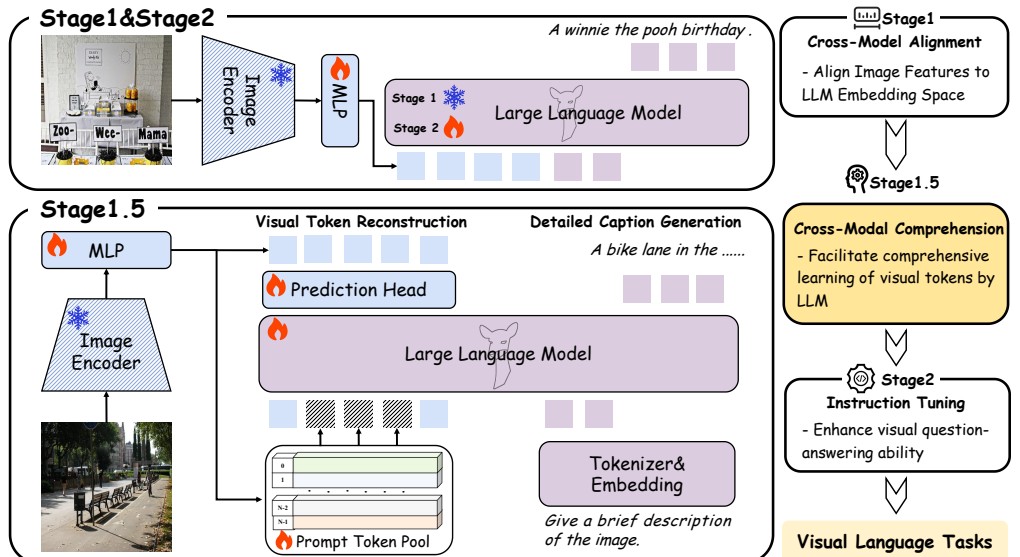

Figure 2: The training pipeline of our proposed Croc model. In contrast to LLaVA-1.5 (Liu et al., 2024a), we introduce an additional pre-training stage that involves novel visual token reconstruction by LLM and targets detailed caption generation. We find that guiding LLM in comprehensive visual token learning is essential for improving cross-modal comprehension.

## 3.2 CROSS-MODAL COMPREHENSION

In this section, we propose a novel cross-modal comprehension pre-training method. Fig. 2 illustrates the training pipeline of the proposed Croc model. Unlike LLaVA, Croc includes an additional pre-training phase between stages 1 and 2. We conceptualize image tokens as a foreign language of the LLM and design a dynamically learnable prompt token pool to replace part of the original visual tokens with the most relevant prompt tokens. To facilitate the image token reconstruction, we design a mixed mechanism of bidirectional visual attention and unidirectional textual attention. Meanwhile, we introduce a detailed caption generation task to further enhance the LLM's understanding of visual tokens.

**Prompt Visual Token Generation.** Inspired by EVA (Fang et al., 2023), we use LLM to reconstruct the masked visual tokens conditioned on visible image tokens. Given an image $I$, we first extract visual features using the image encoder $F_v = E_v(I)$. Following LLaVA, we select the features before and after the last Transformer layer, and the visual projector translates the visual features into visual tokens $T_v = \{v_1, v_2, \ldots, v_n\}$. After that, unlike the previous works (He et al., 2022; Li et al., 2023b), we introduce a learnable prompt token pool to replace part of visual tokens with the most relevant prompt tokens. To ensure full utilization of the prompt tokens in the token pool, we use the Hungarian algorithm (Kuhn, 1955) to associate each masked visual token with a corresponding prompt token. We denote the prompt token pool as $T_p \in \mathbb{R}^{N \times D}$, where $N$ and $D$ represent the number of prompt tokens and the feature dimension. Under the mask ratio of $\gamma$, we get the set of masked visual tokens $\widetilde{T}_v$ awaiting replacement. We pad $\widetilde{T}_v$ with $\varnothing$ into a set of size $N$. To find a bipartite matching between $\widetilde{T}_v$ and $T_p$, we search for a permutation of $N$ elements $\sigma \in \mathfrak{S}_N$ with the lowest cost:

$$\hat{\sigma} = \underset{\sigma \in \mathfrak{S}_N}{\arg\min} \sum_i^N \left( \|\widetilde{T}_v^i - T_p^{\sigma(i)}\|_2 \right) \tag{2}$$

**Mixed Attention Mechanism.** Due to the inherent disparities between visual and textual tokens, the causal interactions between visual tokens are significantly weaker than those observed between textual tokens. Inspired by the UNILM (Dong et al., 2019), we introduce a mixed cross-modal attention mechanism. As illustrated in Fig. 7, to enhance pre-training effectiveness and facilitate a more comprehensive assimilation of visual tokens by the LLM, we employ bidirectional attention among visual tokens and unidirectional attention among textual tokens. Such an arrangement not only allows mutual attention among visual tokens but also allows each textual token to attend to all

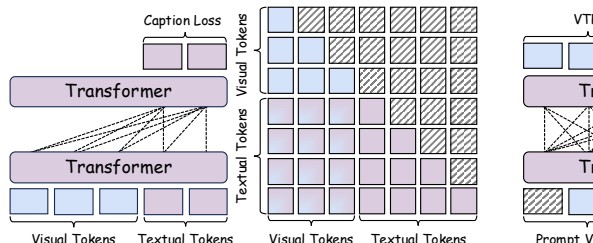
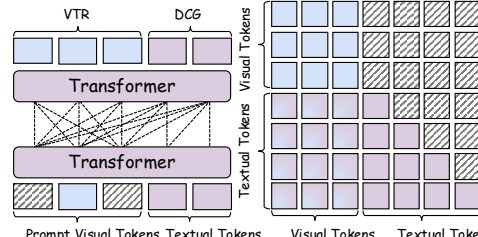

(a) LLaVA Attention Mechanism        (b) Our Mixed Attention Mechanism

Figure 3: Comparisons of cross-modal attention. Different from the unidirectional attention used in LLaVA, our method mixes bidirectional visual attention and unidirectional textual attention. The colored and striped squares represent "attended" and "not attended", respectively.

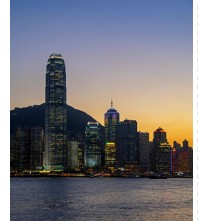

**BLIP Caption:** A large body of water with a city in the background.

**ShareGPT4V Caption:** The image captures a breathtaking view of the Hong Kong skyline at sunset. The sky, awash with hues of orange and blue, serves as a stunning backdrop to the city's architectural marvels. The tallest building, the International Finance Centre, pierces the sky with its imposing height. Its lights are switched on, casting a warm glow that contrasts with the cool tones of the evening sky.\n\nThe other buildings, though not as tall, are no less impressive. They are adorned with lights that twinkle like stars against the twilight sky. The water below mirrors the sky's colors, adding to the overall vibrancy of the scene.\n\nThe perspective of the image is from the water, looking towards the shore. This viewpoint allows for a comprehensive view of the cityscape, from the towering skyscrapers to the smaller structures nestled among them. The image encapsulates the essence of Hong Kong's urban landscape, a blend of modernity and natural beauty.

Figure 4: Comparison of brief caption generated by BLIP (Li et al., 2022) and detailed caption from ShareGPT4V (Chen et al., 2024b). The detailed caption contains rich semantic information of images, which facilitates deep visual token learning by LLM.

visual tokens. Therefore, this attention configuration significantly enhances the efficacy of LLM in understanding and learning from visual tokens.

**Detailed Caption Data.** To improve LLM's understanding of visual tokens, we propose to use detailed caption data for pre-training. As shown in Fig. 4, compared to brief captions, the detailed captions contain a greater wealth of semantic information about image details, thus providing better guidance for LLM to learn visual tokens more effectively.

**Training Objectives.** To improve the large language model's ability to learn visual tokens, we introduce two specific objectives: Visual Token Reconstruction (VTR) and Detailed Caption Generation (DCG). In the prompt visual token generation step, we randomly replace some of the original visual tokens $T_v = \{v_1, v_2, ..., v_n\}$ with tokens from our prompt token pool, thus obtaining the prompt visual tokens $\hat{T}_v$. Then we concatenate the prompt visual tokens $\hat{T}_v$ with the instruction text tokens $T_t = \{t_1, t_2, ..., t_m\}$ and feed them into an LLM to generate a response of length $L$, the probability of generating the contextual response $T_r = \{r_i\}_{i=1}^L$ can be calculated:

$$p(T_r|\hat{T}_v, T_t) = \prod_{i=1}^{L} p(r_i|\hat{T}_v, T_{t,<i}, T_{r,<i}). \tag{3}$$

After receiving the response $T_r$, we split the first 576 tokens $T_{rv}$ to compute the visual token reconstruction loss $\mathcal{L}_{\mathrm{VTR}}$ and the remaining tokens $T_{rt}$ to compute the detailed caption generation loss $\mathcal{L}_{\mathrm{DCG}}$.

The visual token reconstruction loss is calculated as follows:

$$\mathcal{L}_{\mathrm{VTR}} = \sum_{i \in \Theta} \|T_{rv}^i - T_v^i\| \tag{4}$$

where $\Theta$ represents the index set of replaced visual tokens. Meanwhile, we maximize the likelihood of text tokens $T_{rt}$ by employing the auto-regressive language modeling objective:

$$\mathcal{L}_{\mathrm{DCG}} = \sum_i \log p\left(t_i \mid \hat{T}_v, t_1, \cdots, t_{i-1}\right) \tag{5}$$

The overall training loss is the combination of $\mathcal{L}_{\mathrm{VTR}}$ and $\mathcal{L}_{\mathrm{DCG}}$:

$$\mathcal{L} = \alpha \mathcal{L}_{\mathrm{VTR}} + \mathcal{L}_{\mathrm{DCG}} \tag{6}$$

where $\alpha$ is a loss weight used to balance the influence of different losses.

| Method | LLM | Res. | Pretrain | Finetune | VQAv2 | GQA | VizWiz | SciQA-I | TextVQA |
|--------|-----|------|----------|----------|-------|-----|--------|---------|---------|
| BLIP-2 | Vicuna-13B | $224^2$ | 129M | – | 65.0 | 41.0 | 19.6 | 61.0 | 42.5 |
| InstructBLIP | Vicuna-7B | $224^2$ | 129M | 1.2M | – | 49.2 | 34.5 | 60.5 | 50.1 |
| InstructBLIP | Vicuna-13B | $224^2$ | 129M | 1.2M | – | 49.5 | 33.4 | 63.1 | 50.7 |
| Shikra | Vicuna-13B | $224^2$ | 600K | 5.5M | 77.4* | – | – | – | – |
| IDEFICS-9B | LLaMA-7B | $224^2$ | 353M | 1M | 50.9 | 38.4 | 35.5 | – | 25.9 |
| IDEFICS-80B | LLaMA-65B | $224^2$ | 353M | 1M | 60.0 | 45.2 | 36.0 | – | 30.9 |
| Qwen-VL | Qwen-7B | $448^2$ | 1.4B$^\dagger$ | 50M | 78.8* | 59.3* | 35.2 | 67.1 | **63.8** |
| Qwen-VL-Chat | Qwen-7B | $448^2$ | 1.4B* | 50M | 78.2* | 57.5* | 38.9 | 68.2 | 61.5 |
| LLaVA-1.5-7B | Vicuna-7B | $336^2$ | 558K | 665K | 78.5* | 62.0* | 50.0 | 66.8 | 58.2 |
| LLaVA-1.5-13B | Vicuna-13B | $336^2$ | 558K | 665K | 80.0* | 63.3* | 53.6 | 71.6 | 61.3 |
| Croc-7B | Vicuna-7B | $336^2$ | 558K+1.5M | 665K | 80.5* | **64.2*** | 50.0 | 70.1 | 60.4 |
| Croc-13B | Vicuna-13B | $336^2$ | 558K+1.5M | 665K | **80.7*** | 64.0* | **57.1** | **72.7** | 60.8 |

Table 1: Comparison with SoTA methods on academic task oriented datasets. We mark the best performance **bold** and the second best underlined. Croc achieves the best performance on 4/5 benchmarks. * The training images/annotations of the datasets are observed during training.

| Method | LLM | Res. | Pretrain | Finetune | POPE | MME | MMB | MMB-CN | SEED | LLaVA-W | MM-Vet |
|--------|-----|------|----------|----------|------|-----|-----|--------|------|---------|--------|
| BLIP-2 | Vicuna-13B | $224^2$ | 129M | – | 85.3 | 1293.8 | – | – | 46.4 | 38.1 | 22.4 |
| InstructBLIP | Vicuna-7B | $224^2$ | 129M | 1.2M | 86.1 | – | 36.0 | 23.7 | 53.4 | 60.9 | 26.2 |
| InstructBLIP | Vicuna-13B | $224^2$ | 129M | 1.2M | 78.9 | 1212.8 | – | – | – | 58.2 | 25.6 |
| Shikra | Vicuna-13B | $224^2$ | 600K | 5.5M | – | – | 58.8 | – | – | – | – |
| IDEFICS-9B | LLaMA-7B | $224^2$ | 353M | 1M | 81.9 | – | 48.2 | 25.2 | – | – | – |
| IDEFICS-80B | LLaMA-65B | $224^2$ | 353M | 1M | 66.0 | – | 54.5 | 38.1 | – | – | – |
| Qwen-VL | Qwen-7B | $448^2$ | 1.4B | 50M | – | – | 38.2 | 7.4 | 56.3 | – | – |
| Qwen-VL-Chat | Qwen-7B | $448^2$ | 1.4B | 50M | – | 1487.5 | 60.6 | 56.7 | 58.2 | – | – |
| LLaVA-1.5-7B | Vicuna-7B | $336^2$ | 558K | 665K | 85.9 | 1510.7 | 64.3 | 58.3 | 58.6 | 63.4 | 30.5 |
| LLaVA-1.5-13B | Vicuna-13B | $336^2$ | 558K | 665K | 85.9 | 1531.3 | 67.7 | **63.6** | 61.6 | 70.7 | 35.4 |
| Croc-7B | Vicuna-7B | $336^2$ | 558K+1.5M | 665K | 86.9 | 1526.4 | 67.6 | 59.7 | 64.1 | 71.9 | 34.9 |
| Croc-13B | Vicuna-13B | $336^2$ | 558K+1.5M | 665K | **87.8** | **1591.4** | **69.9** | 62.9 | **64.2** | **74.7** | **36.2** |

Table 2: Comparison with SoTA methods on benchmarks for instruction-following LMMs. We mark the best performance **bold** and the second best underlined. Croc achieves the best performance on 6/7 benchmarks.

## 3.3 TRAINING PIPELINE

The overall training pipeline of our Croc model is shown in Fig. 2. Building upon LLaVA-1.5, the Croc undergoes a two-stage pre-training procedure followed by instruction tuning.

**Stage 1: Cross-Modal Alignment.** Following LLaVA-1.5, we first pretrain the projection layer with the identical 558K pretraining dataset used in LLaVA-1.5 to align image features to the LLM embedding space. During training, we keep both the visual encoder and the LLM weights frozen.

**Stage 1.5: Cross-Modal Comprehension.** Building on the cross-modal alignment stage, we introduce the cross-modal comprehension phase as a subsequent pre-training stage. To facilitate comprehensive learning of visual tokens by the LLM, we pretrain the projection layer as well as the LLM in this stage. We select 1.2M detailed image-text pairs from the ShareGPT4V (Chen et al., 2024b) dataset, as the detailed captions (as shown in Fig. 4) can enhance the visual token learning of the LLM. To prevent the degradation of the inherent capabilities of LLM, we also incorporate 300K pure text data. Please refer to the appendix for more details.

**Stage 2: Instruction Tuning.** Following LLaVA-1.5, we freeze the visual encoder weights and update both the pre-trained weights of the projection layer and LLM to improve its visual question answering capabilities. We employ the identical 665K instruction dataset used in LLaVA-1.5.

## 4 EXPERIMENTS

### 4.1 IMPLEMENTATION DETAILS AND EVALUATION BENCHMARKS

Following LLaVA-1.5, we utilize CLIP ViT-L/14@336px (Radford et al., 2021) as the visual encoder and the Vicuna (Chiang et al., 2023) 7b/13B model as the LLM. During the cross-modal alignment stage, the learning rate for the projection layer is set to $1e-3$. In the cross-modal comprehension stage, the learning rates are adjusted to $2e-5$ for both the LLM and projection layer and $1.5e-4$ for the prompt token pool and prediction layer. The random mask ratio $\gamma$ is set as $75\%$ and the size of the prompt token pool is set to $2,048$. In the instruction tuning stage, we adopt the

| Method | LLM | Res. | Pretrain | Finetune | MMStar | DocVQA | AI2D | RealWorldQA |
|--------|-----|------|----------|----------|--------|--------|------|-------------|
| BLIP-2 | Vicuna-13B | $224^2$ | 129M | – | – | – | – | – |
| InstructBLIP | Vicuna-7B | $224^2$ | 129M | 1.2M | – | – | – | – |
| InstructBLIP | Vicuna-13B | $224^2$ | 129M | 1.2M | – | – | – | – |
| Shikra | Vicuna-13B | $224^2$ | 600K | 5.5M | – | – | – | – |
| IDEFICS-9B | LLaMA-7B | $224^2$ | 353M | 1M | 21.6 | – | 42.2 | 42.1 |
| IDEFICS-80B | LLaMA-65B | $224^2$ | 353M | 1M | 26.1 | – | 54.8 | 49 |
| Qwen-VL | Qwen-7B | $448^2$ | 1.4B | 50M | 32.5 | **65.1** | 57.7 | 37.8 |
| Qwen-VL-Chat | Qwen-7B | $448^2$ | 1.4B | 50M | 37.5 | 62.6 | **62.3** | 49.3 |
| LLaVA-1.5-7B | Vicuna-7B | $336^2$ | 558K | 665K | 30.3 | 28.1 | 55.5 | 54.8 |
| LLaVA-1.5-13B | Vicuna-13B | $336^2$ | 558K | 665K | 32.8 | 30.3 | 60.8 | 55.3 |
| Croc-7B | Vicuna-7B | $336^2$ | 558K+1.5M | 665K | 37.7 | 32.5 | 59.3 | 56.3 |
| Croc-13B | Vicuna-13B | $336^2$ | 558K+1.5M | 665K | **37.9** | 33.2 | 59.5 | **57.1** |

Table 3: Comparison with SoTA methods on multimodal understanding benchmarks. We mark the best performance **bold** and the second best underlined.

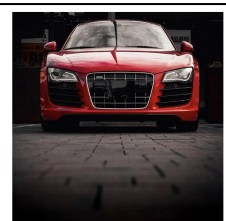

Prompt: <image>\nThe car in the picture appears to be split open. Do you agree with my observation? Please state your judgment and provide reasons.

LLaVA-1.5: Yes, I agree with your observation. The car in the picture is indeed split open, revealing its interior. This is evident from the image, which shows the car's hood and trunk open, exposing the engine and other components inside. The open car door also contributes to the impression of the car being split open.

Croc: No, I do not agree with your observation. The car in the picture is not split open. It is a red Audi car parked on a street, and the image is taken from a low angle, which gives the impression that the car is split open. However, this is just an optical illusion created by the perspective of the photo.

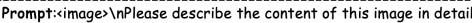

Prompt:<image>\nPlease describe the content of this image in detail.

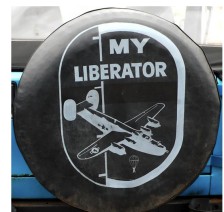

LLaVA-1.5: The image features a black and white sticker on a blue surface, possibly a car or a truck. The sticker depicts an airplane, with the words "My Liberator" written below it. The airplane is prominently displayed, taking up a significant portion of the sticker. The combination of the black and white design and the blue background creates a striking contrast, making the airplane and the words stand out.

Croc: The image captures a moment on a road where a black tire with a white rim is prominently featured. The tire is adorned with a sticker that is a striking contrast of black and white. The sticker is a graphic representation of a military aircraft, specifically a bomber, with the words "MY LIBERATOR" inscribed above it. The aircraft is depicted in a dynamic pose, as if caught in mid-flight. Below the aircraft, there is a parachute. The tire, with its unique sticker, stands out against the backdrop of a blue car, hinting at a story waiting to be told.

Figure 5: Qualitative Comparison of LLaVA-1.5-7B and Croc-7B. Benefiting from our proposed cross-modal comprehension stage, the Croc model exhibits enhanced visual comprehension and reasoning capabilities. Moreover, it can generate enriched image descriptions.

learning rate of $2e-5$ for LLM and the projection layer. We employ AdamW (Loshchilov & Hutter, 2019) as the optimizer, initialized with a weight decay of $0.2$. The parameters $\beta_1$ and $\beta_2$ are set to $0.9$ and $0.98$, respectively. We train Croc on $8\times$ NVIDIA A100 (80G) GPUs.

To prove the effectiveness of the Croc model, we evaluate our models across various benchmarks, including 1) General Visual Question Answering: GQA (Hudson & Manning, 2019), VQAv2 (Goyal et al., 2017b), VizWiz (Gurari et al., 2018), LLaVA-Bench (In-the-Wild) (Liu et al., 2024b), MM-Vet (Yu et al., 2024), RealWorldQA; 2) OCR-Related Question Answering: TextVQA (Singh et al., 2019), DocVQA (Mathew et al., 2021); 3) Illusion Benchmarks: POPE (Li et al., 2023c); 4) Comprehensive Reasoning Benchmarks: MMBench (Liu et al., 2024c), MME (Yin et al., 2023), MMStar (Chen et al., 2024a), MM-Vet (Yu et al., 2024); 5) Science Visual Question Answering: SciQA (Lu et al., 2022), AI2D (Kembhavi et al., 2016);

## 4.2 QUANTITATIVE ANALYSIS

In Tab. 1, we present a detailed comparison between our model and previous state-of-the-art methods across various benchmarks tailored for academic tasks. Croc outperforms existing models on 4/5 benchmarks. Specifically, compared to LLaVA-1.5-7B, Croc-7B demonstrates performance improvements of 2.0%, 2.2%, 3.3%, and 2.2% on VQAv2, GQA, SciQA-I, and TextVQA. Similarly, Croc-13B shows improvements of 0.7%, 0.7%, 3.5%, and 1.1% over LLaVA-1.5-13B on VQAv2, GQA, VisWiz, and SciQA-I. Additionally, we compare Croc with baselines on instruction-following

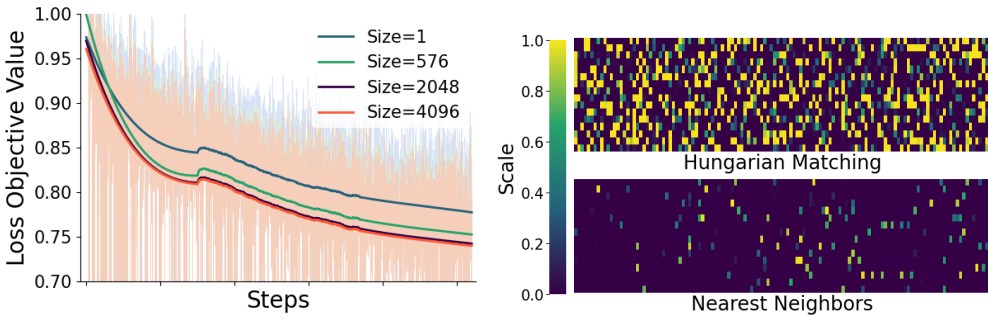

(a) Loss curves with different token pool sizes    (b) Utilization rate of pool tokens

Figure 6: (a) Visualization of the pre-training loss curves for different prompt token pool sizes. Setting the pool size to 1 means that only one learnable token is used to replace all masked visual tokens. (b) The prompt token utilization rate with different matching algorithms. All experiments are based on the Croc-7B model.

| Pool Size | GQA | TextVQA | MMB | SEED | MMStar |
|---|---|---|---|---|---|
| 1 | 62.3 | 56.6 | 65.1 | 61.3 | 34.8 |
| 576 | 63.8 | 58.4 | 66.2 | 62.5 | 33.8 |
| 2048 | **64.2** | **60.4** | **67.6** | **64.1** | **37.7** |
| 4096 | 63.9 | 58.8 | 66.5 | 63.1 | 34.7 |

(a) Ablation on prompt token pool size.

| Mask Ratio | GQA | TextVQA | MMB | SEED | MMStar |
|---|---|---|---|---|---|
| 0.00 | 63.1 | 45.9 | 66.1 | 62.0 | 33.3 |
| 0.50 | 63.8 | 59.2 | 67.6 | 62.6 | 34.6 |
| 0.75 | **64.2** | **60.4** | **67.6** | **64.1** | **37.7** |
| 0.90 | 63.0 | 58.2 | 65.6 | 62.2 | 34.3 |

(b) Ablation on mask ratio.

| Method | GQA | TextVQA | MMB | SEED | MMStar |
|---|---|---|---|---|---|
| Nearest Neighbors | 63.1 | 58.9 | 66.5 | 62.5 | 35.0 |
| Hungarian Matching | **64.2** | **60.4** | **67.6** | **64.1** | **37.7** |

(c) Ablation on matching algorithm.

| Attention Mechanism | GQA | TextVQA | MMB | SEED | MMStar |
|---|---|---|---|---|---|
| Unidirectional Attention | 63.4 | 58.7 | 66.6 | 62.5 | 35.2 |
| Mixed Attention | **64.2** | **60.4** | **67.6** | **64.1** | **37.7** |

(d) Ablation on attention mechanism.

| Caption Type | GQA | TextVQA | MMB | SEED | MMStar |
|---|---|---|---|---|---|
| Brief Caption | 62.5 | 57.1 | 66.4 | 62.9 | 36.1 |
| Detailed Caption | **64.2** | **60.4** | **67.6** | **64.1** | **37.7** |

(e) Experiment results without detailed captions.

| Stage1 | Stage1.5 | GQA | TextVQA | MMB | SEED | MMStar |
|---|---|---|---|---|---|---|
| ✗ | ✓ | 63.2 | 57.7 | 64.7 | 62.1 | 33.1 |
| ✓ | ✓ | **64.2** | **60.4** | **67.6** | **64.1** | **37.7** |

(f) Experiment results without Stage 1.

Table 4: Ablation experiments results. All the experiments are based on the Croc-7B model.

benchmarks. As shown in Tab. 2, Croc achieves superior performance on 6/7 datasets. Notably, compared to LLaVA-1.5-7B, Croc-7B exhibits significant performance gains of 1.0%, 15.7, 3.3%, 1.4%, 5.5%, 8.5%, and 4.4% on POPE, MME, MMB, MMB-CN, SEED, LLaVA-W, and MM-Vet. Furthermore, we evaluate Croc's performance on the multimodal understanding benchmarks in Tab 3. Croc-7B obtains significant improvements of 7.4%, 4.4%, 3.8%, and 1.5% over LLaVA-1.5-7B on MMStar, DocVQA, AI2D, and RealWorldQA, respectively. These performance improvements are primarily attributed to the proposed novel cross-modal comprehension pretraining stage, which significantly enhances the LLM's capability to integrate and understand visual tokens.

## 4.3 QUALITATIVE ANALYSIS

Attributable to the cross-modal comprehension stage, our proposed Croc model adeptly captures fine-grained semantic information from images, significantly enhancing its visual understanding and reasoning capabilities. In the first example shown in Fig. 5, unlike LLaVA-1.5, our model captures more detailed information from images, such as specific vehicle models like Audi and lower shooting angles. As illustrated in the second example of Fig. 5, our Croc model efficiently identifies and describes detailed semantic information, such as the "parachute" under the aircraft.

## 4.4 ABLATION STUDY

**Prompt Token Pool Size.** The prompt token pool is designed to replace part of the original visual tokens with the most relevant prompt tokens. As shown in Fig. 6a, the larger the number of tokens in the pool, the easier it is to generate detail captions, resulting in a lower training loss ($\mathcal{L}_{\text{DCG}}$). The decrease in training loss is saturated when the pool size is 2,048. When the pool size is 1, 75% of

| Method | Data | $\mathcal{L}_{VFR}$ | $\mathcal{L}_{DCG}$ | GQA | SciQA-I | TextVQA | POPE | MME | MMB | MMB-CN | SEED | MMStar |
|---|---|---|---|---|---|---|---|---|---|---|---|---|
| LLaVA-1.5-7B | ✗ | ✗ | ✗ | 62.0 | 66.8 | 58.2 | 85.9 | 1510.7 | 64.3 | 58.3 | 58.6 | 33.3 |
| LLaVA-1.5-7B* | ✓ | ✗ | ✗ | 62.3 | 68.2 | 57.0 | 86.1 | 1450.7 | 62.0 | 53.9 | 60.9 | 33.0 |
| Croc-7B | ✓ | ✓ | ✗ | 62.5 | 68.8 | 56.1 | **87.1** | 1458.5 | 64.3 | 57.6 | 60.4 | 33.9 |
| Croc-7B | ✓ | ✗ | ✓ | 63.3 | **70.5** | 58.5 | 86.9 | 1513.8 | 65.9 | 58.0 | 62.6 | 33.6 |
| Croc-7B | ✓ | ✓ | ✓ | **64.2** | 70.1 | **60.4** | 86.9 | **1526.4** | **67.6** | **59.7** | **64.1** | **37.7** |

Table 5: Ablation experimental results on the additional 1.5M pre-training data and different pre-training objectives. * Results of LLaVA-1.5 we reproduced by adding additional 1.5M pre-training data to the cross-modal alignment stage.

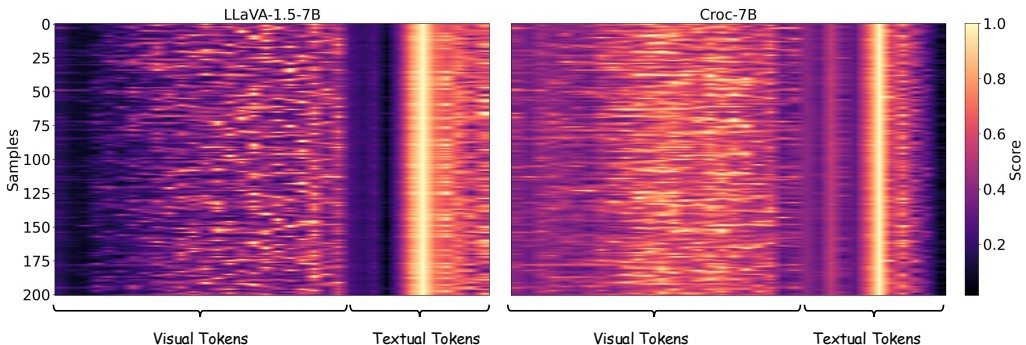

Figure 7: Average of attention scores of language response tokens to visual tokens and language instruction tokens. Here, the same question "Please describe in detail what is in the picture." is used for 200 randomly selected images.

the image tokens are directly dropped and the training is too difficult, resulting in poor performance in Tab. 4a. We observe that increasing the token pool from 1 to 2,048 tokens improves performance. However, expanding the pool further to 4,096 tokens degrades performance because the visual token reconstruction task is too easy to improve the detailed caption generation task.

**Visual Token Mask Ratio.** The mask ratio of visual tokens directly affects the difficulty of the visual token reconstruction task, and thus significantly affects the effectiveness of pretraining. In Table. 4b, we report the results of experiments with different mask ratios. Similar to the observations with MAE (He et al., 2022), we find that using a 75% masking ratio yields optimal results in several downstream benchmarks. Lower ratios, e.g., 0%, make the pretraining task too easy, while higher ratios, e.g., 90%, make the reconstruction task too difficult. Both extremes lead to reduced pretraining effectiveness.

**Nearest Neighbors v.s. Hungarian Matching.** To improve the utilization rate of the tokens in the prompt token pool, we utilize Hungarian Matching to replace 75% of visual tokens with the most relevant prompt tokens. In Fig. 6b, we present a comparative analysis of token utilization using both Nearest Neighbors and Hungarian Matching. Due to the Hungarian Matching algorithm's stringent requirement to select distinct prompt tokens, there has been a significant improvement in the overall utilization of prompt tokens. This enhancement of utilization substantially improves the representational capabilities of the model (Zhu et al., 2024). Therefore, as shown in Tab. 4c, employing the Hungarian Matching algorithm, in contrast to the Nearest Neighbors algorithm, yields substantial performance enhancements across all evaluated benchmarks.

**Unidirectional Attention v.s. Mixed Attention.** To verify the impact of the mixed attention mechanism, we conduct experiments to compare mixed attention with unidirectional attention used in LLaVA-1.5. As shown in Tab. 4d, our proposed mixed attention mechanism achieves significant performance improvement on all the benchmarks. To further explore the influence of different attention mechanisms, we compare the average attention scores of response tokens to visual tokens and language instruction tokens in Fig. 7. Here, the same question "Please describe in detail what is in the picture." is used for 200 randomly selected images. The Croc-7B model exhibits significantly higher attention scores for visual tokens compared to the LLaVA-1.5-7B model, as indicated by the brighter region in the visual token part of the heatmap. This suggests that Croc-7B puts more emphasis on understanding and attending to the image content when generating responses.

| Method | Pretrain | Finetune | VQAv2 | GQA | VisWiz | SciQA$^I$ | POPE | MMB | MMB-CN | SEED | LLaVA-W | MM-Vet |
|---|---|---|---|---|---|---|---|---|---|---|---|---|
| VILA-7B | 50M | 1M | 79.9 | 62.3 | **57.8** | 68.2 | 85.5 | 68.9 | **61.7** | 61.1 | 69.7 | 34.9 |
| Croc-7B | 558K+1.5M | 802K | **80.1** | **63.5** | 55.2 | **72.3** | **86.9** | **69.1** | 60.5 | **63.0** | **73.3** | **36.8** |
| VILA-13B | 50M | 1M | 80.8 | 63.3 | **60.6** | 73.7 | 84.2 | 70.3 | 64.3 | 62.8 | 73.0 | 38.8 |
| Croc-13B | 558K+1.5M | 802K | **80.9** | **64.0** | 57.8 | **74.1** | **87.0** | **71.0** | **65.3** | **64.6** | **80.5** | **39.2** |

Table 6: Performance comparisons between VILA v.s. Croc.

**Brief Caption v.s. Detailed Caption.** To investigate the impact of detailed captions on our proposed cross-modal comprehension stage, we employ the BLIP model to generate brief captions for the 1.2M images from the ShareGPT4V dataset. As shown in Tab. 4e, the use of detailed captions achieves a significant performance improvement. This is mainly because the detailed caption contains rich semantic details of images (as shown in Fig. 4), facilitating visual token reconstruction by LLM.

**Ablation on Stage, Data, and Objective.** In Tab. 4f, we present the performance results without the cross-modal alignment pretraining stage. The misalignment between the features of the LLM and the visual encoder significantly increases the difficulty of the visual token reconstruction task. Consequently, the performance of the model on all the benchmarks is reduced considerably.

In Tab. 5, we present a series of comprehensive ablation studies to elucidate the influence of pretraining data and different training objectives. Initially, we augment the 558K pertaining data of LLaVA-1.5 with the additional 1.5M data and apply the same training method as LLaVA-1.5. This augmentation leads to performance improvements in only 4/9 datasets, while the others exhibit declines. This phenomenon can be attributed to the significant alteration in the distribution of the original 558K pretraining dataset caused by the introduction of the additional data. Building upon this foundation, we incorporated a visual token reconstruction task, which facilitated the establishment of more effective integration between the LLM and visual tokens. This integration led to improved performance across 5/9 datasets. Additionally, benefiting from the detailed description captions, we observe performance improvements in 8/9 datasets after implementing the detailed caption generation task. Finally, by combining visual token reconstruction and detailed caption generation tasks, the detailed caption further enriches the LLM's understanding of visual features, resulting in significant performance enhancements in our model.

**VILA v.s. Croc.** As VILA incorporates an additional one million instruction data, we construct an 802K dataset for the instruction tuning stage to facilitate a more equitable comparison. Specifically, we expand the identical 665K instruction data used in LLaVA-1.5 to 802K by sampling an additional 142K instances from publicly accessible datasets. Please refer to the appendix for more details. As shown in Tab. 6, although our model only utilizes 1/25 of the pre-training data compared to VILA, Croc-7B achieves significant performance improvements in 8/10 of the benchmarks. Similarly, Croc-13B also achieves significant performance improvements in 9/10 of the benchmarks. These performance improvements demonstrate that our proposed cross-modal comprehension pretraining stage can facilitate the learning of visual tokens by LLMs, thereby substantially improving the visual comprehension and reasoning capacities of LMMs.

## 5 CONCLUSION

In this paper, we introduce a novel pretraining paradigm for LMMs to enhance the visual comprehension capabilities of LLMs. Our approach incorporates a new cross-modal comprehension stage designed to bridge the gap between the visual and textual domains. Specifically, we develop a dynamically learnable prompt token pool and apply the Hungarian algorithm to replace a portion of the original visual tokens with the most relevant prompt tokens. To further improve the model's understanding, we propose a mixed attention mechanism that combines bidirectional visual attention with unidirectional textual attention, enabling a more comprehensive interpretation of visual tokens. Additionally, we incorporate a detailed caption generation task, utilizing rich descriptions to improve the LLM's grasp of visual semantic information. Experimental results show that our method achieves state-of-the-art performance across multiple vision-language benchmarks. We hope that our work offers valuable insights for advancing large multimodal models.

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

# A APPENDIX

## A.1 IMPLEMENTATION DETAILS

### A.1.1 HYPERPARAMETERS

In Tab. 7, we present all the training hyperparameters used in different training stages. We use greedy decoding for evaluation to ensure reproducibility.

| Hyperparameter | Stage 1 | Stage 1.5 | Stage 2 |
|---|---|---|---|
| batch size | 256 | 128 | 128 |
| lr | 1e-3 | 1.5e-4/2e-5 | 2e-5 |
| lr schedule | cosine decay | cosine decay | cosine decay |
| lr warmup ratio | 0.03 | 0.03 | 0.03 |
| weight decay | 0 | 0 | 0 |
| epoch | 1 | 1 | 1 |
| optimizer | AdamW | AdamW | AdamW |
| DeepSpeed stage | 2 | 3 | 3 |

Table 7: Hyperparameters used in different training stages.

### A.1.2 DATASETS

In the cross-modal comprehension stage, to prevent the degradation of the inherent capabilities of LLM, we also incorporate 300K pure text data. Fig. 8a shows the composition of a dataset consisting of text-only data. This data includes 123K, 52K, 30K, 25K, 20K, 25K, 15K, and 10K samples from MathInstruct (Yue et al., 2023), Standford Alpaca (Taori et al., 2023), BELLE (BELLEGroup, 2023), OpenPlatypus (Lee et al., 2023), CodeAlpaca (Chaudhary, 2023), Firefly[1], Webqa (Chang et al., 2022), Dolly (Conover et al., 2023).

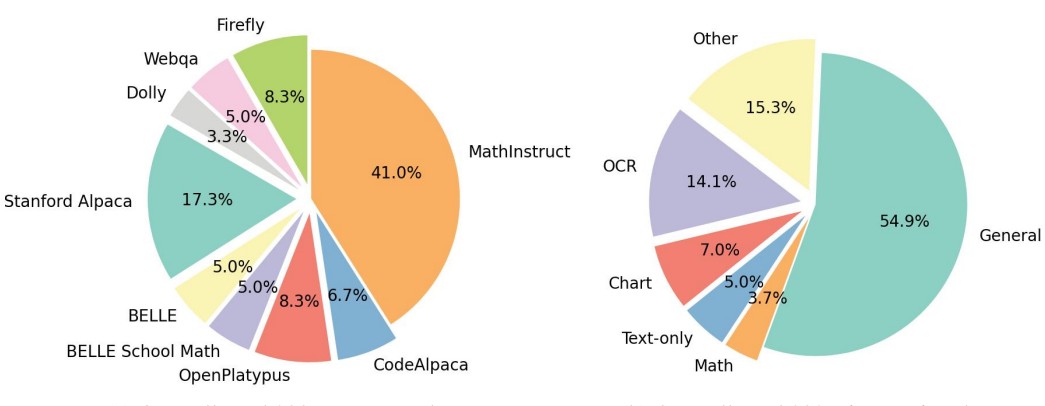

(a) Our collected 300K pure text data.    (b) Our collected 802K instruction data.

Figure 8: Visualization of the proportion of different data in our collected 300K pure text data and 802K instruction data.

To facilitate a more equitable comparison with VILA, we construct an 802K instruction dataset categorized into General, OCR, Chart, Math, Text-only, and Other. Fig. 8b illustrates the distribution of data across various categories. Specifically, the General dataset comprises 158K, 86K, 83K, 82.5K, 11K, 11K, and 9K samples sourced from LLaVA-Instruct (Liu et al., 2024b), VG (Krishna et al., 2017), VQAv2 (Goyal et al., 2017a), A-OKVQA (Schwenk et al., 2022), ShareGPT4V (Chen et al., 2024b), LAION-GPT4V [2], and OKVQA (Marino et al., 2019). The OCR dataset includes 80K, 23K, 5K, 3K, and 2K samples from OCRVQA (Mishra et al., 2019), SynDog-EN (Kim et al., 2022), ST-VQA (Biten et al., 2019), IAM (Cheng et al., 2022), and InfoVQA (Mathew et al., 2022). Chart data is drawn from 10K, 18K, and 28K samples taken from DocVQA (Mathew et al., 2021),

---

[1]https://github.com/yangjianxin1/Firefly
[2]https://huggingface.co/datasets/laion/gpt4v-dataset

---

**Example: Visual Understanding & Reasoning**

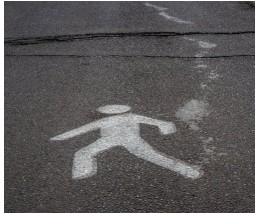 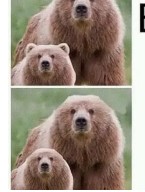 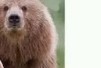

**Prompt:** <image>\nThere seems to be someone walking on the road in the picture. Could you help me guess how fast this person is walking?

**LLaVA-1.5-7B:** Based on the image, it is difficult to accurately determine the speed of the person walking on the road. However, since the person is described as a "pedestrian" and the image shows a white arrow on the road, it is likely that the person is walking at a moderate pace, possibly at a casual or leisurely pace. The presence of the arrow on the road suggests that the area might be designated for pedestrian traffic, which could also influence the walking speed.

**Croc-7B:** It is not possible to accurately determine the speed of the person walking in the picture, as there is no visual information about their movement or the time it takes for them to cross the street. The image only shows a white pedestrian crossing symbol on the road, which is a static representation of a person walking.

**Prompt:** <image>\nWhat's interesting about this picture?

**LLaVA-1.5-7B:** The interesting aspect of this picture is that it features two bears, one large and one small, standing next to each other. The large bear is positioned on the left side of the image, while the smaller bear is on the right side. The image is presented in a split-screen format, with the two bears occupying different sections of the frame. This composition creates a visually striking contrast between the two bears, emphasizing their size difference and capturing the viewer's attention

**Croc-7B:** The interesting aspect of this picture is that there are two bears, one big and one small, at the top of the picture, corresponding to the word 'Bears'; at the bottom of the picture, there are also two bears, one big and one small, but their ears seem to be missing, corresponding to the word 'B'.

Figure 9: Qualitative comparison on visual understanding & reasoning.

ChartQA (Masry et al., 2023), and DVQA (Kafle et al., 2018). To enhance the model logical capacity, we respective sample 5K data from Geomverse (Kazemi et al., 2023), Clevr (Johnson et al., 2017), IconQA (Lu et al., 2021b), RAVEN (Huang et al., 2023), Metamath (Yu et al., 2023) and append additional 3K, 1K, and 1K data from Goat (Liu & Low, 2023), Inter-GPS (Lu et al., 2021a), and LIMA (Zhou et al., 2024). Besides, we append 40K pure text from ShareGPT [3] to prevent the degradation of the inherent capabilities of LLM and append 72K, 48K, and 2.5K other data from GQA (Hudson & Manning, 2019) and RefCOCO (Kazemzadeh et al., 2014), AI2D (Kembhavi et al., 2016).

## A.2 MORE QUALITATIVE RESULTS

We present additional qualitative comparisons on visual understanding & reasoning (Fig. 9), image captioning (Fig. 10), and OCR (Fig. 11) tasks, respectively. Benefiting from our proposed cross-modal comprehension stage, the Croc model exhibits enhanced visual comprehension and reasoning capabilities. Meanwhile, it can accurately generate semantic image descriptions and demonstrate robust OCR capabilities.

## A.3 LIMITATION DISCUSSION

In this work, we have focused solely on image token reconstruction, which limits the scope to static images. However, for comprehensive video understanding, it is essential to consider both spatial and temporal token reconstruction. This would allow us to capture the dynamic changes that occur across frames and enhance the model's ability to process and interpret video sequences more effectively. Expanding our approach to include spatial-temporal token reconstruction is a necessary step for future improvements in video analysis.

## A.4 ETHIC DISCUSSION

Even though large multimodal models represent a major technological advancement with broad applications, their development and deployment bring significant ethical responsibilities. To ensure that the proposed Croc model benefits society while minimizing potential harm, it is essential to address key concerns such as bias, privacy, misinformation, environmental impact, intellectual property, accountability, and human oversight. These challenges are not unique to Croc but are pressing issues faced by the entire AI research and development community.

---

[3]https://huggingface.co/datasets/RyokoAI/ShareGPT52K

## Example: Image Captioning

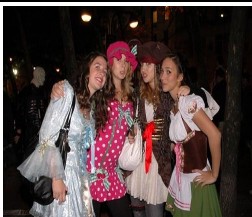 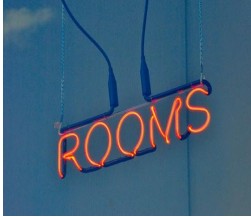

**Prompt:** <image>\nDescribe this image.

**LLaVA-1.5-7B:** The image features a group of four young women dressed in costumes, posing together for a picture. They are all wearing hats, with one of the women wearing a pink hat. The women are standing close to each other, with one on the left, one in the middle, and two on the right side of the frame. Each woman is carrying a handbag, with one handbag located on the left side of the group, another on the right side, and the third one in the middle. The group appears to be enjoying their time together, possibly at a costume party or event.

**Croc-7B:** The image features a group of four women dressed in costumes, posing together for a picture. They are all wearing different outfits, with one woman wearing a pirate costume, another in a fairy costume, and the other two in dresses. The women are standing close to each other, with one woman on the left side, another in the center, and the other two on the right side of the group. There are also a few other people in the background, but they are not the main focus of the image. Additionally, there are two handbags visible in the scene, one near the leftmost woman and the other near the woman in the center.

**Prompt:** <image>\nPlease describe the content of this image in detail.

**LLaVA-1.5-7B:** The image features a neon sign that reads "rooms" in a bright orange color. The sign is hanging from a ceiling, possibly in a hotel or motel setting. The sign is positioned in the middle of the scene, with the word "rooms" being clearly visible. The orange color of the sign stands out against the background, making it easy to read and notice.

**Croc-7B:** The image presents a scene dominated by a neon sign that boldly announces "ROOMS". The sign, glowing with a vibrant orange hue, is affixed to a blue wall, creating a striking contrast. The sign itself is rectangular, that frames the orange text. It's suspended from the wall by two blue wires, one on each side, giving the impression that it's floating in mid-air. The sign is slightly tilted to the left, adding a touch of dynamism to the otherwise static scene. The background is a dark blue, which further accentuates the neon sign and the blue wall. The overall composition of the image suggests it might be a sign for a hotel or a similar establishment.

Figure 10: Qualitative comparison on image captioning.

## Example: OCR

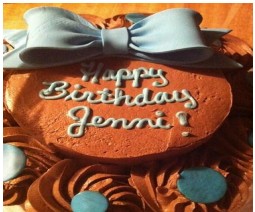 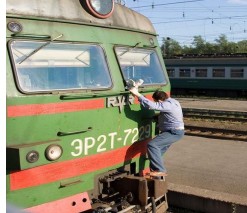

**Prompt:** <image>\nWhat's on the cake in the picture?

**LLaVA-1.5-7B:** The cake in the picture has a blue bow on top, and it is decorated with chocolate frosting.

**Croc-7B:** The cake in the picture has a blue bow and the words "Happy Birthday Jennie" written on it.

**Prompt:** <image>\nWhat are the numbers on the locomotive?

**LLaVA-1.5-7B:** The numbers on the locomotive are 3P2-720.

**Croc-7B:** The numbers on the locomotive are 3P2T-7229.

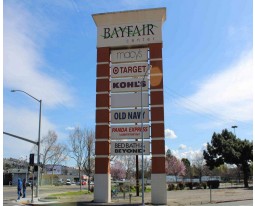 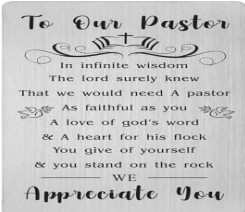

**Prompt:** <image>\nPlease identify the words in the picture.

**LLaVA-1.5-7B:** Bay Fair Mall.

**Croc-7B:** The words in the picture are: BAYFAIR center, Macy's, TARGET, KOHL'S, OLD NAVY, PANDA EXPRESS (Located in Food Court), BED BATH & BEYOND

**Prompt:** <image>\n Please identify the words in the picture.

**LLaVA-1.5-7B:** To our pastor, in infinite wisdom, the Lord surely knew, that we would need a pastor, as faithful as you, a love of God's word and a heart for his flock. You give of yourself, you stand on the rock, we stand with you. We appreciate you.

**Croc-7B:** The words in the picture are: To Our Pastor, In infinite wisdom, The lord surely knew, That we would need A pastor, As faithful as you, A love of god's word, & A heart for his flock, You give of yourself, & you stand on the rock, WE, Appreciate You.

Figure 11: Qualitative comparison on OCR.

