# OpenReview forum: "Croc: Pretraining Large Multimodal Models with Cross-Modal Comprehension"
_ICLR.cc/2025/Conference — ICLR 2025 Conference Withdrawn Submission_

### Official Review · Reviewer_fHjW · 2024-10-21

**Soundness:** 3
**Presentation:** 3
**Contribution:** 3
**Rating:** 6
**Confidence:** 4

**Summary:**

This paper mainly proposes a new pre-training paradigm for visual language, aimed at enhancing the joint learning of vision and language during the pre-training phase. The implementation involves adding a cross-modal learning task between stages 1 and 2 based on LLaVA1.5, which includes the reconstruction of visual tokens and the generation of text dense caption tasks. Additionally, full attention is applied to visual tokens to promote the model's understanding of visual features.

**Strengths:**

1) This paper introduces the reconstruction of visual tokens in visual language models, which is somewhat innovative.

**Weaknesses:**

1) The baseline is too low; it may be worth trying better baseline models to validate the results.

2) The scope of the visual token reconstruction effect is a bit limited; could it be extended to the generation of visual signals?

**Questions:**

1)  Based on the current training paradigm, have you tried the effect of removing visual token reconstruction?

---

### Official Review · Reviewer_TvfR · 2024-10-31

**Soundness:** 2
**Presentation:** 2
**Contribution:** 2
**Rating:** 3
**Confidence:** 4

**Summary:**

This paper presents Croc, a MLLM that built upon the shoulder of LLaVA-1.5, with learnable mask token pool, mixed attention (UniLM style) mechanism, and trained with additional detailed caption data.
Croc proposes a new cross-modal comprehension stage to enhance the visual comprehension capability of MLLMs by the above three modifications.
Croc achieves better performance than the original LLaVA-1.5 on most of the downstream tasks.

**Strengths:**

1. This paper proposes to use a learnable mask token pool to replace the visual token, instead of using a fixed learnable mask token used in most MIM models, to ease the difficulty of masked visual representation regression.
2. This paper conducts a comprehensive ablation study to verify the effectiveness of each proposed component or design choice.
3. This paper provides visualizations and qualitative analysis to help understand the model's behavior.

**Weaknesses:**

1. The novelty of this paper is limited. The main contribution of this paper is the proposed cross-modal comprehension stage, which consists of learnable mask token pool, mixed attention mechanism, and additional detailed caption data. However, these components are not novel and have been used in previous works. The combination of these components does not bring new insights.
2. The experimental comparison with LLaVA-1.5 is not convincing. In Figure 1(b), Table 1,2,3, authors compare Croc with the original LLaVA-1.5. However, the comparison is not fair because Croc uses additional 1.5M detailed caption data for pre-training and additional costs bring by the proposed visual token regression loss, masked token pool, and mixed attention mechanism. Notably, authors provide the comparison with LLaVA-1.5 + 1.5M in Table 5, but add 1.5M detailed caption data to stage-1 (projection layer tuned) instead of stage-1.5 / 2 (projection layer and LLM tuned).
3. This paper lacks analysis of the computational cost of the proposed Croc compared to LLaVA-1.5. The additional visual token regression loss, masked token pool, and mixed attention mechanism will bring non-negligible computational costs. The authors should provide a detailed analysis of the computational cost of Croc compared to LLaVA-1.5 in both training and inference stages.

**Questions:**

1. Croc uses additional learnable mask token pool with the Hungarian algorithm to ease the difficulty of masked visual representation regression. However, as authors mentioned, "When the pool size is 1, 75% of the image tokens are directly dropped and the training is too difficult", which means the mask ratio should be less than 0.75 for pool size = 1. Could authors provide the performance of Croc with pool size = 1 and different mask ratios, to provide a stronger evidence for the necessity of the learnable mask token pool?
2. In Table 5, the last four rows are all trained with additional 1.5M detailed caption data. If so, why is L_dcg disabled in the second and third rows?
3. In Table 1,2,3, it is weird that the performance of Croc with 7B sometimes similar (VQAv2, SEED, MMStar, AI2D) to or even better (GQA) than Croc with 13B. Could authors provide some insights into this phenomenon?
4. In Table 6, VILA does not share the same LLM backbone with Croc (LLaMA2 vs Vicuna), authors should add this detail to this table.

---

### Official Review · Reviewer_YhKU · 2024-11-04

**Soundness:** 2
**Presentation:** 2
**Contribution:** 2
**Rating:** 3
**Confidence:** 4

**Summary:**

The paper introduces visual token reconstruction as an additional pretraining task for multi-modal large language model pretraining. To train on the reconstruction task, the proposed method randomly replaces a subset (75% by default) of the visual encoder output tokens with their best non-replacement match from a learnable pool of features, feeds them into a fine-tuned large language model, and then minimizes their distances to the corresponding features before replacement. This training is placed in a separate stage between visual feature alignment and visual instruction tuning, hence named stage 1.5, together with the regular next-token prediction loss. The uni-directional attention is also replaced with a mixed attention with bi-directional attention on the visual tokens. The model is evaluated on a wide range of VQA and MLLM benchmarks.

**Strengths:**

* The general idea of adding feature reconstruction loss for MLLM pretraining is somewhat reasonable.

**Weaknesses:**

* **The term *cross-modal comprehension* is too broad and vague**: The phrase, which is supposed to reflect the core idea and motivation of the proposed method, appears in multiple important places throughout the paper (including main title, section titles, main contributions), yet its specific meaning seems hard to decipher: Every multi-modal model should be capable of cross-modal comprehension by its name. It is even more confusing given that the proposed visual token reconstruction task is arguably closer to intra-modal (i.e., visual-to-visual) modeling, because they cannot see the rich text captions directly due to the causal mask.

* **The motivation of adding token pool is unclear**: It seems that no clear explanation is given for the introduction of the token pool other than adjusting the reconstruction difficulty (Ln 429-460). In such case, it would be necessary to show that it cannot be replaced by straightforward difficulty control methods such as the mask ratio (i.e., what are the results with no token pool and a lower mask ratio, e.g., 50% or 25%?)

* **Comparison fairness with LLaVA-1.5 with additional data**: Since ShareGPT4V is usually used as a post-training dataset, and is used in Croc with the language model unfrozen, it would make another interesting baseline to see what comes out if ShareGPT4V is added to the 2nd stage of LLaVA-1.5 training.

**Questions:**

* **Overhead of bipartite matching**: The Hungarian algorithm is O(N^3) in complexity and has to run serially with the training steps due to data dependency (i.e., it needs the vector pool after the previous update step and generates part of the input for the current forward step, from what is described in Sec. 3.2). What is the training time overhead brought by the Hungarian algorithm? Is any technique used to reduce the overhead? Is a GPU implementation used for it?

* **Mixed attention**: Out of the 3 stages (1, 1.5 and 2), in which stage is mixed attention (i.e., bi-directional attention on visual tokens) added? Does it cause convergence issues when LM is frozen?

---

### Official Review · Reviewer_54bq · 2024-11-04

**Soundness:** 3
**Presentation:** 3
**Contribution:** 3
**Rating:** 6
**Confidence:** 4

**Summary:**

This paper introduces a pretraining paradigm for Large Multimodal Models (LMMs) to enhance the visual comprehension capabilities of Large Language Models (LLMs). The approach includes a cross-modal comprehension stage, where a dynamically learnable prompt token pool replaces part of the original visual tokens using the Hungarian algorithm. Using a mixed attention mechanism (from UNILM), combining bidirectional visual attention with unidirectional textual attention, to improve visual token understanding. Additionally, a detailed caption generation task is integrated to further enrich visual semantic comprehension. Experimental results demonstrate that the proposed model, named Croc, achieves good performance across various vision-language benchmarks.

**Strengths:**

1. The methodology is well-explained, with clear steps and justifications for the chosen techniques, providing a strong foundation for reproducibility.

2. The dynamically learnable prompt token pool, combined with the Hungarian algorithm for selecting relevant tokens, represents an innovative approach to integrating visual information with textual inputs.

3. Unlike previous methods that often freeze certain model parameters to stabilize training, this paper proposes a multi-stage pretraining approach. The multi-stage pretraining method looks made that the visual encoder's parameters remain learnable throughout the process, which can enhance the model's ability to adapt and refine its visual comprehension capabilities over time.

**Weaknesses:**

1. Although the paper is generally clear, some sections, particularly those detailing the implementation of the Hungarian algorithm and the mixed attention mechanism, could benefit from additional explanations or visual aids to further enhance understanding.
To support reproducibility, the paper could provide more comprehensive implementation details.

2. The paper lacks insightful analysis regarding the impact of the multi-stage pretraining approach on the generated model. Reviewers might be curious about how each pretraining stage contributes to the overall performance and understanding of the model. It is noted that in the final Stage 2, the vision part does not have a loss, which raises the question of whether the vision part should only have a supervised loss during the pretraining stages. If the supervised loss is only applied during pretraining, it begs the question of why the process is divided into Stages 1 and 1.5. Could other methods, such as Exponential Moving Average (EMA), achieve similar effects?

**Questions:**

None

---

### Official Review · Reviewer_UzPz · 2024-11-09

**Soundness:** 3
**Presentation:** 3
**Contribution:** 2
**Rating:** 5
**Confidence:** 4

**Summary:**

This paper proposes a new pre-training paradigm for enhancing the visual comprehension capabilities of LLMs. It introduces a novel cross-modal comprehension stage that uses a dynamic learnable prompt token pool and replaces some of the original visual tokens with the most relevant prompt tokens using the Hungarian algorithm. The paper also proposes a detailed caption generation task to help LLMs understand visual semantic information better. Experimental results show that the proposed Croc model achieves state-of-the-art performance on massive vision-language benchmarks.

**Strengths:**

1. The paper is well structured and reads good.
2. The prompt visual token generation with a learnable prompt token pool with Hungarian algorithm is designed to enhance cross-modal comprehension. The mix attention mechanism also helps to bring the improvement.
3. The ablation study is good to show the contribution of each design.

**Weaknesses:**

1. The method is not that novel, since adding detailed caption knowledge, and visual token reconstruction has also been introduced in other works such as LLaVA-OneVision, EVA. Adding an additional pre-training stage with another 1.5M data is natural to improve the performance. I am wondering how is the performance if no further additional 1.5M data is introduced.
2. The authors should compare with more recent state-of-the-art methods such as LLaVA-OneVision, QwenVL-2 and mPLUG-Owl3.
3. More visualization or analysis should be given to what prompt token pool learn and how is the relation between it and the masked vision token.

**Questions:**

1. What is the value of α in Equ.(6)? Which one is more important for the VTR and DCG loss?
2. How is the performance if no further additional 1.5M data is introduced?
3. Can you also give the ablation result on VQA dataset in Table 4?

---

### Note · Authors · 2024-11-14

I have read and agree with the venue's withdrawal policy on behalf of myself and my co-authors.